# Validation of the Health Index in the Postoperative Period: Use of the Nursing Outcome Classification to Determine the Health Level

**DOI:** 10.3390/healthcare12080862

**Published:** 2024-04-20

**Authors:** Sara Herrero Jaén, Alexandra González Aguña, Marta Fernández Batalla, Blanca Gonzalo de Diego, Andrea Sierra Ortega, María del Mar Rocha Martínez, Roberto Barchino Plata, María Lourdes Jiménez Rodríguez, José María Santamaría García

**Affiliations:** 1Mejorada del Campo Health Centre, Community of Madrid Health Service (SERMAS), 28840 Madrid, Spain; 2Research Group MISKC, Department of Computer Science, University of Alcala, Polytechnic Building, University Campus, Barcelona Road Km. 33.6, 28805 Alcalá de Henares, Spain; alexandra.gonzalez@uah.es (A.G.A.); martafernandez@codem.es (M.F.B.); blanca.gonzalo@salud.madrid.org (B.G.d.D.); andrea.sierra@salud.madrid.org (A.S.O.); roberto.barchino@uah.es (R.B.P.); lou.jimenez@uah.es (M.L.J.R.); chesantgar@hotmail.com (J.M.S.G.); 3Santa Cristina University Hospital, Community of Madrid Health Service (SERMAS), 28009 Madrid, Spain; 4Meco Health Centre, Community of Madrid Health Service (SERMAS), 28880 Madrid, Spain; 5Illustrious Official College of Nursing of Madrid, Av. de Menéndez Pelayo, 93, Retiro, 28007 Madrid, Spain; mar.rocha@codem.es; 6Computer Science Department, University of Alcala, 28805 Madrid, Spain

**Keywords:** continuity of care, health services research, health status, nursing care, post-anesthesia nursing

## Abstract

Background: The postoperative period is the recovery time after surgery and is defined as an individual process whose purpose is to return the person to the state of normality and integrity that they had prior to surgery. Aim: Demonstrate the modification of the level of health of people in the early postoperative period through the development and validation of the Health Index Instrument, which is built from the Nursing Outcomes Classification (NOC) standardized language. Design: The design used a mixed method, which involved a first phase of instrument development and a second phase of instrument validation. Methods: The methods was based on focus group techniques with text analysis techniques, internal validation with a group of care language experts, external validation with a group of clinical nursing experts and a clinical validation with quantitative and qualitative analysis. A panel of experts in Language of Care evaluated the (NOC) labels and their correlation with the 11 Health Variables to construct the instrument. The instrument developed was subjected to external validation with a panel of clinical nurse experts in post-anesthesia care. The clinical validation included a cross-sectional descriptive study in a postoperative unit. The final sample of the cross-sectional descriptive study was 139 cases. Results: Of the 89 NOCs proposed in the preliminary construction phase of the instrument, 36 passed through the first round. Of those 36 NOCs, 25 passed through to the second round with a review performance and 11 directly as approved. The total number of approved NOCs were 4. The results of the research show that there are changes in the global score of the health level and in each health variable. It is observed that there was a significant increase in the scores of the health variables at admission and discharge (*p* < 0.001). Conclusions: The results of the data analysis show that six groups present a similar pattern of evolution of the health variables. A correlation was found between the time of stay in the unit with the scores obtained in the health variables, the physical functioning, comfort status and the presence of symptoms being particularly significant.

## 1. Introduction

The World Health Organization defines major surgery as “any procedure performed in the operating room that involves the incision, excision, manipulation, or suturing of tissue, and generally requires regional or general anesthesia or deep sedation to control pain” [1].

Therefore, surgery can be understood as one of the main therapeutic methods directed towards people with health problems [2] that exists to address and provide a solution or palliation to certain health problems [3] for diagnostic, treatment or rehabilitation purposes [4].

The surgical process is constituted by the perioperative period [5], which begins when the need for surgery is identified in a person due to a health problem. The perioperative period is made up of three phases: preoperative (information to the person, acceptance of the surgical process and prior preparation), intraoperative (surgical intervention) and postoperative (recovery after surgical intervention) [6].

The postoperative period is the recovery time after surgery and is defined as an individual process whose purpose is to return to the person’s state of normality. The return of normality means that people return as close as possible to their pre-surgical condition, compatible with survival and related to the health problem that motivated the intervention. In addition, it means that they return to the integrity that they had prior to surgery through the recovery of physical aspects, psychological, social for activities of daily living [7,8,9].

This recovery period can be classified into three stages: early recovery (from the end of the surgery until discharge from the post-anesthesia care unit), intermediate (from now until hospital discharge) and finally late (until recovery of normal functioning of the person) [7,10,11].

The early recovery phase is also known as the immediate postoperative period [11,12], being in this phase in which the vital reflexes of the person are recovered [8]. It is a critical and complex process within the surgical process because, in this short period of time, the greatest number of complications arising from the surgery and the health status of the person appear [13,14].

During this phase, people are at risk for critical health-care needs such as hypoxemia, airway obstruction, hypotension, agitation, nausea and vomiting, pain, or hypothermia [13,15], in addition to those associated with the secondary effects derived from general anesthesia [16,17,18].

General anesthesia is characterized by the reversible reduction of the functions of the Central Nervous System, with a complete loss of perception of all the senses: sensory, mental, motor and reflex blockade [17,19]. Thus, the surveillance and care of people in this vital situation must be maintained until “respiratory autonomy, circulatory balance and neurological status are restored” [20], as well as the control of signs and symptoms, recovering the main vital functions.

To know the existing evidence base related to how to measure health recovery in the postoperative period of the person, a scoping review was performed in PubMed and Scopus with the search phrase (“post operative” [Title]) AND (“scale” [Title]) AND (“scale” [Title]).

The inclusion criteria for the articles were that the date of publication was within the last five years and the language was in English or Spanish. The result of the scoping review was 18 articles referring to 16 different scales.

Among the scales were: the visual analogue scale (VAS), the Detroit Interventional Pain Assessment Scale, Clinical Frailty Scale, Glasgow coma scale, among others. It was identified that 35.95% of the scales were referred to pain, 12.50% to neurological function, 12.50% to frailty and the rest of them to diagnostic aspects or detection of signs such as bleeding. None of those identified used standardized language of care.

According to the World Health Organization, more than 234 million surgical procedures are performed worldwide each year for a wide range of health problems [21].

It should be noted that nursing is the profession in charge of leading the population care [22], in all stages and vital situations in which the person finds themselves.

The American Society of PeriAnaesthesia Nurses (ASPAN) defines the principles of safety and quality in nursing practice in the perioperative and perianesthetic setting [23,24].

In this sense, nurses play an important role in caring for people who require surgery [25]. Specifically, the post-anesthesia care unit (PACU) has highly trained nurses [13] for the care of people to optimize safety in the recovery or maintenance of the level of health. In this sense, applying D. Orem’s model, the nurse is responsible for acting as a partial or total compensatory system [26] in relation to the person’s self-care capacity throughout the post-surgical recovery process.

González et al. developed a theoretical model called the Knowledge Model about Person Care, in which care is a continuum over time, and establishes the person as a central element of care, characterizing them from their vulnerability [22]. This model establishes the health of the person on the one hand as a result of their care and, on the other hand, as a resource to carry out their care in the next moment [22,27,28]. A health status can be represented from health variables (Appendix A), which are a set of 11 variables that cover the different dimensions of health [29].

On the other hand, the nursing discipline has professional models of care [22]. In this way, the nursing profession has several standardized languages for professional practice. Among others, there is the Nursing Outcomes Classification (NOC) [30], a language for the outcome criteria of caring for the person. Previous studies have shown that, although NOC results are not designed for evaluation, its indicators can be used since they represent people’s potential states, including health status [31]. Furthermore, the Spanish legislative framework has Royal Decree 1093/210, of September 3, which approves the minimum set of data for clinical reports in the National Health System, which describes the set of data that nursing must include in the clinical report, among which nursing results are collected through the NOC taxonomy [32].

No specific instruments were found to identify variations in people’s level of health in the postoperative period.

None of the instruments that partially measured any aspect of health found used standardized nursing language.

Thus, this study aimed to demonstrate the modification of people’s health level in the early postoperative period, through the development and validation of the Health Index Instrument, which is constructed from health variables and the standardized NOC language.

## 2. Materials and Methods

### 2.1. Design

The design of this study was a mixed-method study, which involved a first phase of instrument development and a second phase of instrument validation (Figure 1).

The study complied with the Good Reporting of a Mixed-Methods Study (GRAMMS) checklist [33].

The study period was from September 2016 to June 2019.

The procedure applied a mixed methodology where a panel of experts performed the validation through the Delphi technique.

In the context of this study, the definition of expert is based on the model of care defined by Patricia Benner, in which an expert is defined as a person who has an intuitive understanding of the situation and is able to identify the problem.

In addition, an expert demonstrates clinical and resource-based practice mastery, possesses materialized practical knowledge, has a vision of the whole and has the ability to prevent the unexpected. As a result, an expert has the ability to recognize patterns, rules, models or ontologies, thanks to his or her deep experience [34].

The mixed methodology combined focus group techniques with text analysis techniques and an internal validation through consensus by a group of care language experts with a Delphi technique. Then, an external validation was carried out through consensus by a group of clinical experts and clinical validation through the development of a descriptive cross-sectional study to validate the instrument in the postoperative resuscitation unit of the Severo Ochoa University Hospital.

The design was based on other instrument development studies, which involved a development phase and an evaluation phase of the instrument [35,36,37].

This combination of methods, from the initial design of the instrument to its validation, allowed the benefits of the different techniques to be obtained, as well as reducing the limitations and possible biases of using a single method.

### 2.2. Development Phase

The structure of the tool is based on the health variables set out in the Introduction, which describe the elements set out in the World Health Organization’s definition of health as follows (Appendix A).

Having established this structure and for the construction of the instrument, the Nursing Outcomes Classification (NOC) taxonomy was selected for its operationalization, so that a qualitative and numerical value, and therefore a measurement, could be assigned to each of the health variables. This allows the final calculation of a Health Index.

Preliminary questions were developed based on the definitions and attributes of the concepts presented through the concept analysis [35].

#### 2.2.1. Construction of a Preliminary Instrument

This phase aimed to construct the preliminary instrument, and it was based on determining the relationships between the health variables and the Nursing Outcomes Classification (NOC) taxonomy [30].

In this theoretical phase of the study, health variables and NOC labels of the Nursing Outcomes Classification (NOC) taxonomy are considered as participants. The selection criteria for this taxonomy were because, in the first instance, it formulates health outcome goals to be achieved by the individual that can be assigned a score (from 1 to 5) in relation to their corresponding scale.

One researcher elaborated the preliminary construction of the instrument: a woman expert in care language and in postoperative clinical care with a master’s degree, PhD student and more than eight years of research and clinical experience.

For the development of the instrument, firstly, preliminary questions were developed based on the definitions and attributes of the health variables and their definition presented through the concept analysis [35]. The preliminary questions were as follows:Does the nursing outcome include in its label any words found in a health variable or in its definition?Does the nursing outcome include in its definition any word found in any health variable or in its definition?Does the nursing outcome include in its indicators any words found in a health variable or its definition?

Therefore, on the one hand, there were health variables and their definition, and on the other hand, there were the NOC labels, their definition and their indicators.

The methodology used is based on the research carried out by González et al. [38].

The expert nurse analyzed the health variables and their definition, which were written in natural language. These variables were translated into their equivalents in standardized NOC taxonomy language.

The expert nurse then analyzed the NOC taxonomy and extracted those NOC outcomes that shared equivalent concepts for each health variable. These concepts can be found in the NOC label itself, in its definition or in its indicators.

The search sequence for establishing that there was a correlation between health variable and NOC label was derived from these matching criteria:There had to be coincidences of common concepts between the definition of the health variable and the definition of the NOC label.Or, there had to be a conceptual coincidence (synonymous) between the definition of the health variable and the indicators of the NOC label.

These criteria are necessary as they are two different areas: A global institution (WHO) and a professional discipline.

In this way, the total number of NOC outcomes that met the inclusion criteria were selected. This selection followed heuristic reasoning because it chose the most relevant NOC label to represent the specific meaning of each health variable [38].

#### 2.2.2. Internal Validation: Validation by Experts in the Language of Care

This phase aimed to validate the proposed relationships between health variables and NOC labels. The selection of NOC labels obtained in the previous phase were presented to a group of care language experts to validate the relationship between each health variable to a single NOC label and thus to its respective scale. This conceptual relationship was carried out into successive phases until a 1-to-1 relationship was obtained, which means that one health variable was related to one NOC label.

The panel was composed of seven care language experts (Table 1). The participants of the expert panel were selected through purposive sampling. The selection applied the following criteria for a person to be considered an expert: more than five years of work experience in various fields related to the nursing profession, more than five years of experience in the use of languages of care, more than five years of experience in the development of different knowledge-based systems, have at least a master’s degree of academic level, and currently working as a nurse in one of the following recognized profiles (clinical, management, teaching or research).

None of the panel participants had been involved in the research at any previous stage. All participants were introduced to the project and given a briefing, and they also signed a confidentiality agreement prior to participating in the study.

A total of 7 participants were selected based on convenience, and all of them met the eligibility criteria as experts, based on previous studies. Six of the experts had clinical experience (both in Primary Care and Critical Care), and seven had postgraduate studies and research experience. All of them were researchers of the MISKC Research Group (Research in Informatics in Health Care), an interdisciplinary research group [39], and members of the Madrid Scientific Society of Care (SoCMaC) [40]. The characteristics of the experts formed a group of specialists in the research field with the capacity to make a consensus judgement on the research.

The study applied a quantitative validation using the Delphi technique. To carry out the Delphi, a questionnaire was developed in a table format. The first column included the health variables. The second column included the definition of the variable. The third column included the NOC labels selected in the preliminary development phase of the tool. The fourth column included the definition of each NOC. The fifth column included the indicators included in each label and NOC, and finally, the last column included the score estimated by the expert for the evaluation of the relationship between the health variable and the NOC. It was used with a 3-point Likert scale (high rating = 3 medium rating = 2 and low rating = 1) [41,42]. Each expert evaluated the outcome under the question: Do you consider this NOC outcome appropriate for this health variable?

Each NOC included the number of responses on a Likert scale. An NOC was a “pass” when all the experts gave a score of 3. The NOCs that have passed on to the second round were those whose score was a pass and to which the majority of experts (5/8) gave a score of 2; therefore, they were considered to have a good assessment and advanced to the second round [35,43].

The NOCs that passed through the second round were resubmitted to the experts until a single NOC was obtained by consensus of all the experts.

The consensus criterion was that the same NOC label should represent as many health variables as possible (criterion of maximum representativeness), thus reducing the cost of clinical information necessary for the use of the instrument.

Two rounds were necessary, which were temporally separated by two days. After the two rounds, a consensus criterion was achieved to obtain the ratio of 1 to 1, which means that one health variable was related to one NOC label.

The second part of the validation carried out a qualitative study through a focus group discussion with all the experts of the panel. The content of the session was the presentation of the results that were obtained, and all the experts agreed with them.

### 2.3. Validation Phase

This phase was carried out in two consecutive phases. The first step was to obtain external validation of the instrument developed in the previous phase within the scope of application of the instrument. This was performed by a group of clinical experts in postoperative care nursing. The second step consisted of a clinical validation obtained through a descriptive cross-sectional study with the aim of verifying its applicability in clinical practice.

#### 2.3.1. External Validation

External validation was the first phase, performed by a group of clinical nursing experts who will subsequently participate in the second phase in the data collection, in the clinical validation phase. The experts were recruited using the convenience technique and applying eligibility criteria based on clinical and academic merit [44]. None of the panel participants had participated in any previous phase of the research.

The clinical nurse expert group consisted of 5 clinical nurse experts in post-anesthesia care (3 women and 2 men). All members had at least ten years of experience in clinical care, experience in the use of tools for the assessment and evaluation of health situations and had different clinical and academic profiles. As a clinical profile, the group included one supervisor in a post-anesthesia critical care unit and four with a critical care nursing profile. From the academic profile, one had a master’s degree, one had a law degree (as well as a nurse) and the other participants had postgraduate training.

All participants received a presentation session of the project. For the group of clinical experts, a total of 2 face-to-face sessions of 1 h each were held, with an interval of two days between each session. For validation, the instrument obtained in the development phase was presented to the team of experts. The question to validate each health variable and its correlation with the NOC label was “Do you consider that this Health Variable and its correlation with the NOC label allows to determine the level of health of a person?” with the possible answers being limited to “yes”, “no” and “no answer”. In the second session, the “think aloud” technique was used [45].

#### 2.3.2. Clinical Validation

The last phase included the development of a cross-sectional descriptive study to validate the instrument. The setting for this study was a public university hospital in the Community of Madrid; specifically, in the Postanesthesia Care Unit of the Severo Ochoa University Hospital (Community of Madrid). The unit has a total of 12 clinical management units with 24 h coverage. Of the total of 12 clinical management units, 6 have resources to care for patients who may have critical health problems and 6 have resources to care for people who have undergone rapid recovery surgery and whose stay in the unit will be of a shorter duration.

According to the latest report published in 2018, scheduled surgical activity at this hospital is around 13,000 surgeries per year, of which 10,000 are outpatient and 3000 require hospitalization [46]. According to data provided by the hospital, a total of 12,575 surgeries were scheduled in 2019.

From January to June 2019, the principal investigator and the five clinical nurse experts collected the data. All of them signed a confidentiality document. In addition, they were trained in the use of the instrument through face-to-face sessions and mock cases.

The study was carried out over four distinct periods spanning four months each to reduce any potential seasonal biases and to limit the alteration of results. Also, the summer period was excluded because the permanent staff of the unit was on holiday. The fieldwork was carried out from Monday to Friday, the days when most of the scheduled surgical interventions take place. Saturdays, Sundays and public holidays were excluded because only urgent surgeries were performed on those days.

A purposive sample was carried out under the following inclusion and exclusion criteria:Inclusion Criteria: 18 years of age or older, signature of informed consent, scheduled surgery and general anesthesia.Exclusion criteria: Under 18 years of age, patients incapable of giving consent, people with cognitive impairment, urgent surgeries, local and regional anesthesia, being admitted to the unit before the start of each study cut-off, continuing to be admitted to the unit after the study cut-off and remaining in the unit for more than 7 h.

The criteria for establishing the cut-off at a unit stay of 7 h or less is because the working day is a 7 h shift (8:00 a.m. to 15:00 p.m. or 15:00 p.m. to 22:00 h). In this way, the possibility of alteration in data collection was reduced by limiting the number of experts to a maximum of two per case studied (if the patient remained in the unit for both shifts).

##### Sample and Data Collection

The diagram for obtaining the final study sample is shown in Figure 2.

For the data collection phase, a questionnaire was developed based on the validated health index measurement instrument. The descriptive variables of the study were collected from the person’s demographic data, which were included in the computerized medical history in the Selene^®^ computer program: age, sex and type of surgery. On the other hand, the health variables were the variables that measured the study factors.

The questionnaire had to be completed at least twice by each participant (on admission to the unit immediately after surgery and before discharge). In addition, it was to be carried out every hour or whenever there was a significant change in the participant’s health status, always recording the time of realization.

##### Ethical Considerations

This study has been reviewed and approved by the ethics committee of the Severo Ochoa University Hospital, where we conducted the study dated (14 December 2018), before data were collected.

The confidentiality and secrecy of personal information were respected, following the data protection law in force in Spain, according to Regulation (EU) 2016/679 of the European Parliament and of the Council of 27 April 2016 on Data Protection (GDPR) [47].

To ensure the confidentiality of personal data, each participant was assigned an anonymous numerical code, with no personally identifiable information. In addition, all participants were asked to sign an informed consent form and received written and oral information about the study. They were informed that participation was voluntary and that they could drop out of the study at any time.

##### Data Analysis

The data collected were analyzed by descriptive statistics using measures of central tendency and dispersion (mean, range, percentage) grouped in frequency distribution tables and categorical data classified (health variables, sex, age, surgery).

We used as a reference manual for data analysis the book entitled *Nursing Research: Generating and Assessing Evidence for Nursing Practice* [43] and the book *Statistics Applied to Health Sciences* [48].

Data were processed and analyzed with R Studio program [49]. We checked for data entry errors. Descriptive statistics were used to assess participant characteristics and item data.

To ensure the validity of the data, prior to the study, we conducted a pilot (n = 30) in which the data collector researchers used the instrument in the same cases in order to compare health outcomes and to ensure that response criteria were the same for each researcher. In this way, we avoided researcher bias in data collection. The method used to analyze the data was based on studies conducted within the MISKC Research Group [39].

In the study, two types of health variables were found:Contextually significant variables: Physical Functioning, Mental Functioning, Social Functioning, Comfort State, Presence of Signs, Presence of Symptoms, Physical Condition, Mental Condition, Social Condition.Contextually non-significant variables: material resource and time resource. This was because the study participants were hospitalized in a health institution. The National Health System in Spain is free and universal [50] so it was assumed that they had the necessary material resources for their care, as well as all their time to take care of themselves.

## 3. Results

### 3.1. Instrument Construction

The results obtained for the internal validation of the relationship of the health variables with the NOC labels are shown in Appendix A. All experts on the panel completed the first and second rounds. Of the 89 NOCs proposed in the preliminary phase of instrument construction, 36 passed through to the first round. Of those 36 NOCs, 25 advanced to the second round with a revision interpretation, and 11 passed through directly as approved.

In the second round, the 36 NOCs that passed through were presented again to the experts, until a single NOC was obtained by consensus of all experts for each variable. The total number of NOCs approved was 4 (the same NOC label could be on more than one health variable, but each health variable could only have one NOC label).

In addition, the experts did not add other possible NOCs for any health variables and reviewed the results that were not validated in the first round.

This resulted in the following relationship: one health variable with one NOC, and therefore with its scale, thus allowing a score from 1 to 5 to be assigned to each health variable (1 being the worst score and 5 the best score). The Health Index is defined as the result of the sum of the scores of health variables.

### 3.2. External and Clinical Validation

#### 3.2.1. External Validation

External validity is determined by the clinical acceptability of the developed theoretical construct by experts in the field of the study. Given the context of application of the instrument, acceptability is established by means of an absolute dichotomous variable with the response options “yes” or “no”.

The five clinical experts who made up the panel in the external validation answered “yes” to the question “Do you consider that this Health Variable and its correlation with the NOC label allows to determine the level of health of a person?” for all health variables.

In addition, they were given the opportunity to present allegations or proposals, and none were obtained.

#### 3.2.2. Clinical Validation

Clinical validity was determined by correlating the value obtained by the instrument with the standardized discharge criteria of the Postanesthesia Care Unit. The correlation between the Health Index and the Postanesthesia Care Unit discharge criteria was similar for all cases studied. Both the Health Index value and the unit’s discharge criteria were standardized according to the baseline of the person’s normal state prior to the health problem.

As for the cross-sectional descriptive study carried out for the clinical validation of the instrument, the final study sample was comprised of 139 cases, of which 92 were women (66.18%) and 47 men (33.81%).

The characteristics of the population are representative of those seen in the hospital where the study was conducted. Participants were aged between 20 and 86 years (SD 15.85). The mean age of the participants was 54.4 years for men and 53.5 years for women.

In relation to the surgical specialty, 42.44% of the cases correspond to general and digestive surgery, followed by gynecology with 20.14%, 12.23% with traumatology, 7.91% with urology, 10.79% with otorhinolaryngology and 6.47% with vascular surgery of the total sample. The distribution according to sex and surgical specialty is shown in Figure 3.

For data analysis, the sample was separated by sex. This separation by sex is justified by the fact that the distribution and predisposition to present certain health problems is different in relation to the sex of the person. In this sense, the World Health Organization determines that men have a lower life expectancy, and this is due, among other factors, to those related to gender [51]. Furthermore, there is research that establishes that sex is a conditioning variable in people’s care and, together with other variables, such as the perception of gender limitations, determines their level of vulnerability [28,52].

Therefore, health systems need to take into account gender-related differences by incorporating a gender perspective in order to take the necessary health-care measures to ensure equity in people’s health care.

In addition, the sample was divided into six groups according to the time spent in the unit. Group 1 consisted of those study subjects who were admitted to the unit for 1 h, representing 7.91% of the total sample. Group 2 were those subjects who were hospitalized for 2 h, representing 42.44% of the total sample. Group 3 were those subjects who were hospitalized for 3 h, which amounted to 28.77%. Group 4 were those who stayed for 4 h, accounting for 10.07% of the total sample. Group 5 consisted of those subjects who were hospitalized for 5 h, accounting for 7.19% of the total sample, and finally, Group 6 consisted of those subjects who were hospitalized for 7 h, accounting for 3.59% of the total sample. No study subject was admitted for 6 h.

##### Health Level Evolution

The results of the research show that there are changes in the overall health index score and thus in the health variables.

In addition, it was observed that the rate of recovery of the participants’ level of health is higher in the first two hours after surgery. This is due to the fact that the greatest number of complications occur in the first hours of the immediate postoperative period, as well as their resolution through different therapeutic measures.

Thus, the specific analysis of the level of health by length of stay in hours identified that the average recovery rate in both sexes shows an upward trend in the first hours, remaining stable in the last hours (Figure 4) (Table 2).

##### Differential Health Level between Admission and Discharge

The differential between admission and discharge for each health variable is presented as grouped in frequency distribution tables (Table 3). It can be seen that there was a significant increase in health variable scores at admission and discharge in both men and women.

In terms of the Health Index, the modification can be observed at the individual level, but it also occurs harmoniously in the total sample as a whole, being objectified in the radials of the circumference of the graph (Figure 5). The figure shows the Health Index value at admission and discharge per person and sex, graphically representing all the individuals studied, as well as the overall differences in health linked to sex.

In this regard, men’s health at discharge varied on average by 4.2 points with respect to the level of health at admission. Using the health score at admission to and discharge from the postoperative care unit, a significant difference was demonstrated between the two time points (t = −7.8184, df = 90.877, *p* < 0.001).

On the other hand, women’s health status varied on average by 3.6 points. Using the health score at admission to and discharge from the postoperative care unit, a significant difference was demonstrated between the two time points (t = −12.183, df = 168.25, *p* < 0.001).

## 4. Discussion

This study makes it possible to establish a positive health indicator (Health Index) in the face of health alteration processes.

In Europe, the Statistical Office of the European Union collects the “Percentage of people with good or very good perceived health by sex”. This indicator is a subjective measure of how people judge their overall health on an instrument from “very good” to “very bad” [53]. In addition, the European Statistics of Income and Living Condition survey contains a small module on health, consisting of three variables on health status, representing the so-called European Minimum Health Module: self-perceived health, chronic morbidity and activity limitation—disability [54]. However, these variables do not collect information on all dimensions of health [7].

The Health Index is also a proposal for an intermediate health outcome for the person in the process of discharge from hospital. Having intermediate health results, in the hospital setting, will allow establishing intermediate stages in the clinical processes linked to nursing care.

In addition, the Health Index is established as an indicator applicable in other care processes based on the Person-Centered Care Model [22] as well as its implementation in clinical simulation systems or the representation of health situations in technological environments such as digital twins.

Numerous scales and instruments are used to assess the immediate post-surgical recovery of individuals [20,55,56,57]. Currently, health status assessment focuses on the control of signs and symptoms [58,59,60]. Thus, the assessment of pain, mobility, respiratory dynamics, level of consciousness and vital signs control are among the assessment items of several instruments (VAS instrument, Bromage instrument, Glasgow Coma instrument, Ramsay instrument, Steward instrument) [56,57]. Among the most widely used instruments to assess patient recovery is the Aldrete scale, which dates back to 1970, but nowadays we find multiple adaptations [61]. This instrument includes assessment criteria for respiratory status, circulatory status, color, state of consciousness and activity, pain, limb mobility and fluid intake and output.

In 2010, the Postoperative Quality Recovery Survey (PQRS) instrument was developed [62], which performs an objective assessment of postoperative recovery. This instrument assesses the domains of physiological, nociceptive, functional, cognitive and emotional recovery, as well as the patient’s overall recovery [63] but it is not based on professional nursing models.

In that sense, the Health Index instrument incorporates the standardized NOC language. This taxonomy is internationally recognized by the American Nurses Association, the Unified Medical Language System, complies with ISO/TS 18104: 2014 and is included in the international interoperability standards Health Level-7 [38], a language of care used in this study in the multidisciplinary assessment of people’s level of health.

The fact that an instrument incorporates an international standardized language is something new, since any instrument that has been found uses it to measure the people level of health.

Thus, by correlating health variables with a standardized language of health outcomes, the interpretation bias is resolved as it is a known, recognized and accepted language worldwide [30,32]. In this way, it has been shown that in the same patient, there is concordance in the results obtained by different researchers.

As it is a standardized language, it can be implemented in health information computing systems, as well as in technological environments such as Knowledge-Based Systems and Artificial Intelligence.

The collation of the realized results by the different experts aims to ensure the quality of the findings by means of a mixed-method instrument that brings together different perspectives (primary care and critical care). In addition, the techniques include quantitative evaluations, which allows us to better understand the experts’ evaluation [64].

On the other hand, Nilsson et al. confirmed four dimensions of the postoperative recovery process: physical, psychological, social and habitual [7]. The Health Index is a tool that implicitly incorporates these dimensions through the health variables.

A person’s health is considered a fixed resource at a given point in time, and at the same time, it is considered variable over time [22,29]. Face validity means that the tool should show the changes known and expected by health professionals. The study revealed that the instrument had good sensitivity to change in the participants’ health variable scores throughout their post-surgical recovery.

The variables Physical Functioning, State of Comfort and Presence of Signs and Symptoms were the variables with the most significant change in score. This is because the greatest number of complications occur in the first hours of the immediate postoperative period: general malaise, nausea, vomiting, severe pain or anxiety [20,63,65].

There were no significant changes in the variables of physical condition, mental condition and social condition over time. This is because this study did not consider the previous health situation of the individuals and only analyzed what is derived from the surgical process. Regarding mental condition, it should be noted that one of the exclusion criteria for the study was people with cognitive impairment.

In terms of limitations, this study uses a cross-sectional methodology; therefore, it was not possible to apply the test–retest, which would have provided results on the stability of the measures over time.

## 5. Conclusions

The study shows that regardless of the type of surgery performed under general anesthesia, the entire sample shows variation in health variable scores throughout their admission to the postoperative resuscitation unit.

The results of the data analysis show that the total of the six groups present a similar pattern of evolution of the health variables. A correlation was found between the length of stay in the unit with the scores obtained in the health variables, with physical functioning, state of comfort and presence of symptoms being particularly significant.

The instrument is valid for any moment in a person’s life, as well as for any health process, and can be used in future research, as well as in different areas of management and care. It could also be aimed at the evolution and monitoring of the health status of people with chronic health processes, repercussions of the use of medicines on the level of health, resource forecasting, case management or continuity of patient care, among others.

In addition, it is useful for identifying situations in which the health status of the person decreases, or for establishing situations with a higher level of vigilance by health professionals.

Finally, this study shows that a method has been validated that makes it possible to determine the state of health at a given moment in time and may therefore make it possible to describe in the future the evolution of the level of health throughout a person’s life.

The limitations of the study are that the clinical validation of the instrument has been limited to persons admitted to a specific Postanasthesia Care Unit. As an instrument, it would be advisable to extend this validation to other hospitalization settings as well as to Primary Health Care levels in order to increase its clinical applicability.

On the other hand, as an international standardized language was used, no limitations were identified with regard to the measurement indicator. However, its comprehensiveness could be improved by using the ICNP (International Classification of Nursing Practice) language standard [66].

The usefulness of this instrument for measuring health has been demonstrated in the clinical field, and it can be opened up to the field of management, teaching and research itself, from which it arises.

## Figures and Tables

**Figure 1 healthcare-12-00862-f001:**
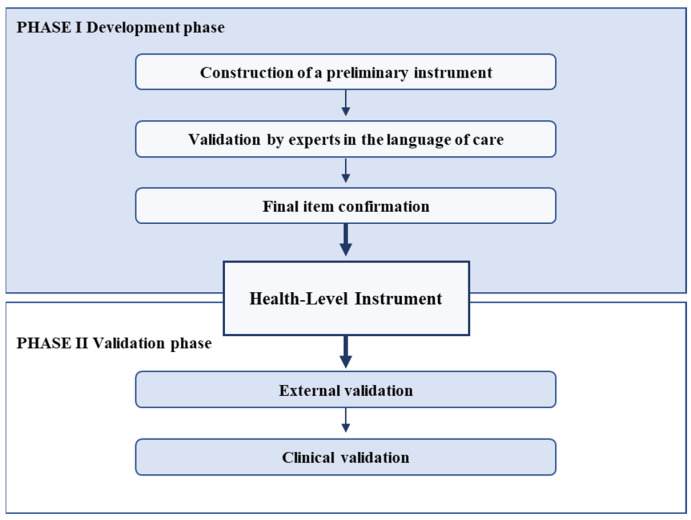
Methodological process.

**Figure 2 healthcare-12-00862-f002:**
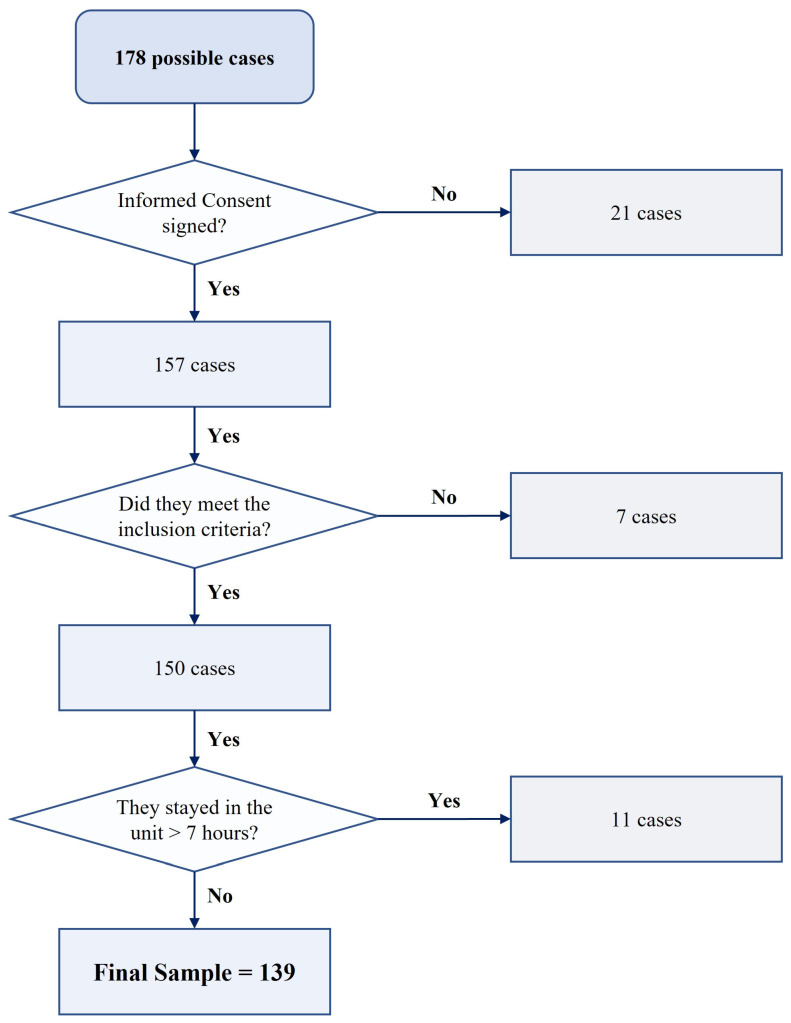
Diagram for obtaining the final sample of the study.

**Figure 3 healthcare-12-00862-f003:**
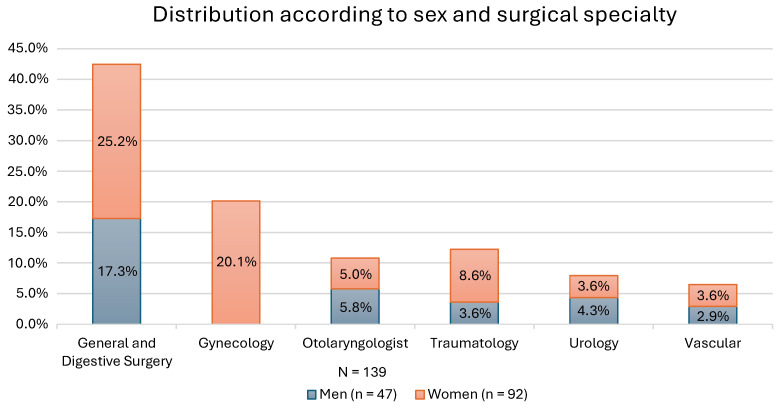
Distribution according to sex and surgical specialty.

**Figure 4 healthcare-12-00862-f004:**
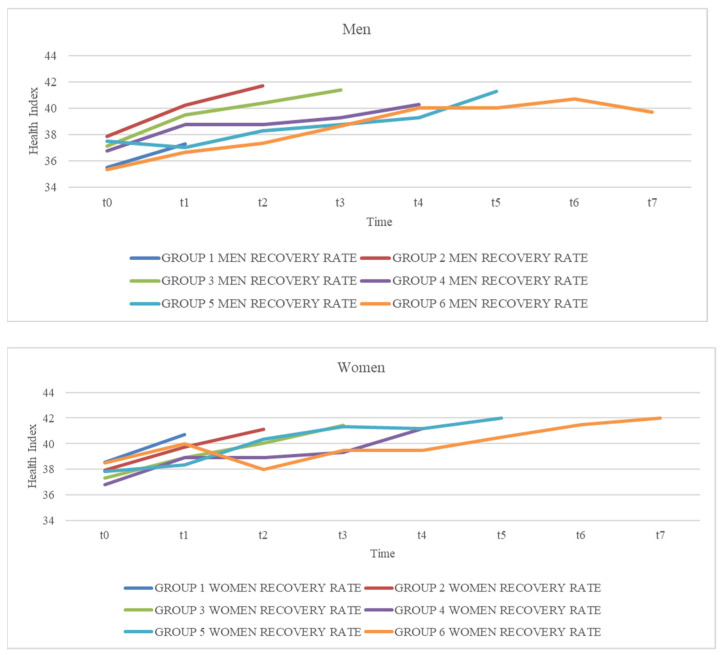
Health recovery index by time groups and sex.

**Figure 5 healthcare-12-00862-f005:**
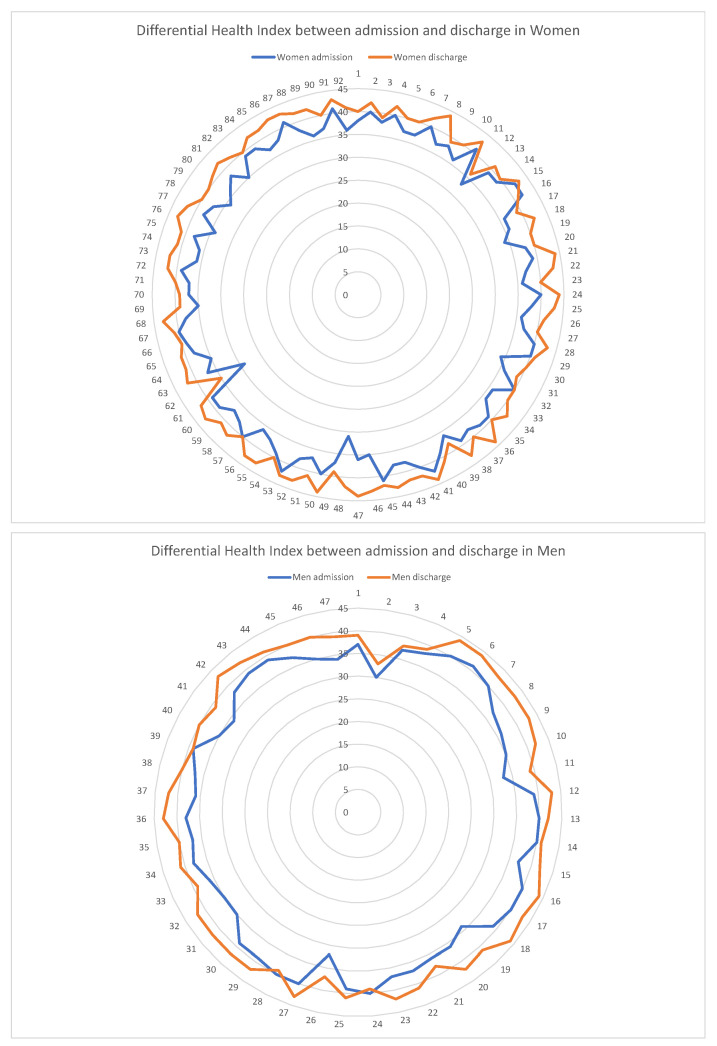
Differential Health Index between admission and discharge.

**Table 1 healthcare-12-00862-t001:** Characteristics of experts in languages of care.

Characteristics (Total Responding)	N	(%)
Gender (N = 7)		
Male	3	42.85
Female	4	57.14
Age (N = 7)		
20–29	2	28.57
30–39	3	42.85
>40	2	28.57
Years of Nursing Practice (N = 7)		
1–10	5	71.42
11–20		
21–30	2	28.57
Years of Experience in the use of Languages of Care (N = 7)		
1–10	5	71.42
11–20	0	0
21–30	2	28.57
Clinical Position (N = 7)		
Clinical nurse	6	85.71
University teaching	1	14.28
Level of Nursing Education (N = 7)		
Graduate	7	100
Master	7	100
Specialist	2	28.57
PhD	2	28.57
Research Experience (N = 7)		
Master	7	100
Doctorate	2	28.57
Research group	7	100
Clinical research	6	85.71

**Table 2 healthcare-12-00862-t002:** Health recovery index by time groups and sex (t = time).

Group and Sex	t_0_	t_1_	t_2_	t_3_	t_4_	t_5_	t_6_	t_7_
Group 1 men recovery rate	35.5	37.25						
Group 1 women recovery rate	38.57	40.71						
Group 2 men recovery rate	37.83	40.20	41.66					
Group 2 women recovery rate	37.94	39.74	41.14					
Group 3 men recovery rate	37.10	39.50	40.37	41.37				
Group 3 women recovery rate	37.31	38.93	40.03	41.43				
Group 4 men recovery rate	36.75	38.75	38.75	39.25	40.25			
Group 4 women recovery rate	36.80	38.90	38.90	39.30	41.20			
Group 5 men recovery rate	37.50	37.00	38.25	38.75	39.25	41.25		
Group 5 women recovery rate	37.83	38.33	40.33	41.33	41.16	42.00		
Group 6 men recovery rate	35.33	36.66	37.33	38.66	40.00	40.00	40.66	39.66
Group 6 women recovery rate	38.50	40.00	38.00	39.50	39.50	40.50	41.50	42.00

**Table 3 healthcare-12-00862-t003:** Distribution of frequencies of the scores of the health variables at admission and discharge by sex (fi: Absolute frequency; fri%: relative frequency).

Health Variables	Score scale	Women				Men			
		Admission		Discharge		Admission		Discharge	
		fi	fri%	fi	fri%	fi	fri%	fi	fri %
Physical Functioning	1	0	0.00%	0	0.00%	0	0.00%	0	0%
	2	1	1.08%	0	0.00%	0	0.00%	0	0.00%
	3	41	44.56%	3	3.26%	21	44.68%	2	4.25%
	4	50	54.34%	84	91.30%	26	55.31%	45	95.74%
	5	0	0.00%	5	5%	0	0.00%	0	0.00%
Mental Functioning	1	0	0.00%	0	0.00%	0	0.00%	0	0.00%
	2	0	0.00%	0	0.00%	1	2.12%	0	0.00%
	3	12	13.04%	0	0.00%	6	12.76%	1	2.12%
	4	69	75.00%	13	14.13%	35	74.46%	5	10.63%
	5	11	11.95%	79	85.86%	5	10.63%	41	87.23%
Social Functioning	1	0	0.00%	0	0.00%	0	0.00%	0	0.00%
	2	0	0.00%	0	0.00%	1	2.12%	0	0.00%
	3	7	7.60%	0	0.00%	3	6.38%	0	0.00%
	4	46	50.00%	4	4.34%	22	46.80%	5	10.63%
	5	39	42.39%	88	96%	21	44.68%	42	89.36%
Comfort Status	1	1	1.08%	0	0.00%	0	0.00%	0	0.00%
	2	4	4.34%	0	0.00%	1	2.12%	0	0.00%
	3	30	32.60%	3	3.26%	19	40.42%	1	2.12%
	4	52	56.52%	59	64.13%	24	51.06%	32	68.08%
	5	5	5.43%	30	32.60%	3	6.38%	14	29.78%
Signs Presence	1	0	0.00%	0	0.00%	0	0.00%	0	0.00%
	2	0	0.00%	0	0.00%	0	0.00%	0	0.00%
	3	11	11.95%	1	1.08%	5	10.63%	0	0.00%
	4	61	66.30%	28	30.43%	33	70.21%	19	40.42%
	5	20	21.73%	63	68.47%	9	19.14%	28	59.57%
Symptoms Presence	1	0	0.00%	0	0.00%	0	0.00%	0	0.00%
	2	1	1.08%	0	0.00%	0	0.00%	0	0.00%
	3	17	18.47%	2	2.17%	13	27.65%	0	0.00%
	4	66	71.73%	61	66.30%	33	70.21%	26	55.31%
	5	8	8.69%	29	31.52%	1	2.12%	21	44.68%
Physical Condition	1	0	0.00%	0	0.00%	0	0.00%	0	0.00%
	2	0	0.00%	0	0.00%	0	0.00%	0	0.00%
	3	6	6.52%	7	7.60%	8	17.02%	8	17.02%
	4	53	57.60%	50	54.34%	25	53.19%	26	55.31%
	5	33	35.86%	35	38.04%	14	29.78%	13	27.65%
Mental Condition	1	0	0.00%	0	0.00%	0	0.00%	0	0.00%
	2	0	0.00%	0	0.00%	0	0.00%	0	0.00%
	3	0	0.00%	0	0.00%	0	0.00%	0	0.00%
	4	7	7.60%	8	8.69%	6	12.76%	6	12.76%
	5	85	92.39%	84	91.30%	41	87.23%	41	87.23%
Social Condition	1	0	0.00%	0	0.00%	0	0.00%	0	0.00%
	2	0	0.00%	0	0.00%	0	0.00%	0	0.00%
	3	0	0.00%	0	0.00%	1	2.12%	1	2.12%
	4	2	2.17%	2	2.17%	0	0.00%	0	0.00%
	5	90	97.82%	90	97.82%	46	97.87%	46	97.87%

## Data Availability

The data that support the findings of this study are available on request from the corresponding author.

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
