# Peer review of "Validation of the Health Index in the Postoperative Period: Use of the Nursing Outcome Classification to Determine the Health Level"

_healthcare, 2024, doi:10.3390/healthcare12080862_

Round 1
Reviewer 1 Report
Comments and Suggestions for Authors
Thank you for providing the opportunity to revise the manuscript titled "Validation of the Health Index in the postoperative period: Use of the Nursing Outcome Classification to determine the health level." The topic is intriguing as it delves into demonstrating the modification of health levels through the Health Index.
However, the manuscript lacks a robust guideline to follow, such as the Mixed Methods Reporting in Rehabilitation & Health Sciences (MMR-RHS).
While the aim of the study is well described and the decision to use a mixed methods approach is justified, there are areas that require improvement.
Overall, the manuscript is too lengthy. I recommend removing certain tables, such as Table 1 and Table 3, and relocating them as supplementary files.
In the Materials and Methods section, the mention of the design being based on other studies lacks specific citation of the theories supporting the methodology, instead only referencing those studies.
Is there a specific reason why the authors have reported the age and gender of linguistic experts?
Tables are lacking of notes and explanations (e.g., Fi, Fri). Moreover i suggest the authors to decide to use one, two or no decimals when reporting numbers.
Figure 5 is not clear for the readers.
Additionally, while two books are mentioned as statistics reference manuals, it is unclear if any data processing programs were utilized. Mentioning a reference manual for data analysis is unusual.
Comments on the Quality of English Languagenone
Reviewer 2 Report
Comments and Suggestions for Authors
Abstract: It is crucial for the authors to provide clearer elucidation regarding the research design and methodology employed in each phase of the study. A lack of explicit detail regarding the methodology leaves readers questioning the robustness and validity of the research process.
Introduction: Why were nursing theories for post-anaesthetic care listed and what does it have to do with the minimum data summary and the NOC? The importance of the study, the state of the art and the potential contributions are unclear
Materials and Methods: The manuscript would significantly benefit from a more comprehensive elucidation of several key methodological aspects pertaining to the development and validation of the Health Index Instrument. Firstly, it is essential to clarify the structural framework of the tool, providing insights into its construction, components, and operationalization. Additionally, readers would greatly benefit from a detailed explanation regarding whether a preliminary systematic review was conducted to inform the development process, shedding light on the existing evidence base and guiding the selection of relevant health variables. Moreover, elucidating the criteria for expert selection and defining the term "expert" within the context of this study is imperative to establish the credibility and validity of the instrument's development process. Furthermore, a clear distinction between participants and inclusion criteria is warranted to avoid ambiguity and ensure the accurate interpretation of study findings. Clarifying these methodological nuances would enhance the transparency and rigor of the research process, thereby strengthening the credibility and utility of the Health Index Instrument within clinical practice and scholarly discourse.
Please, clarify how the internal, external and clinical validity of the study was measured.
Results: The manuscript would greatly benefit from a thorough clarification regarding how the obtained results correspond to the stated objectives of the study and align with the underlying research design. Furthermore, elucidating why the results were presented based on self-defined gender is crucial to ensure transparency and mitigate potential biases. By providing a rationale for this approach, such as acknowledging the significance of individual gender identity in healthcare contexts or addressing specific research questions related to gender disparities, the authors can enhance the interpretability and applicability of their findings. This clarification would not only strengthen the methodological transparency of the study but also facilitate a more nuanced understanding of the implications of the results within the broader context of gender-informed healthcare research.
Discussion: The manuscript would benefit significantly from aligning the discussion of the results more closely with the study's objectives and research design. Such alignment is essential to ensure that the interpretation of findings directly addresses the research questions posed and provides meaningful insights into the study's aims. Additionally, the inclusion of other instruments for assessing the clinical state after anesthesia in the discussion warrants clarification. While the introduction introduced theoretical frameworks relevant to post-anaesthetic care, the inclusion of various assessment instruments in the discussion could provide a comprehensive overview of the current landscape in this domain. However, to maintain coherence, it's imperative to clearly articulate how these instruments contribute to the broader understanding of postoperative recovery and how they complement or differ from the Health Index Instrument under study. By doing so, the discussion can effectively contextualize the significance of the results within the existing literature and offer valuable insights into the practical implications for clinical practice and future research endeavors.

Moderate editing of English language required.
